# Lipidomic Profiling Reveals Biological Differences between Tumors of Self-Identified African Americans and Non-Hispanic Whites with Cancer

**DOI:** 10.3390/cancers15082238

**Published:** 2023-04-11

**Authors:** April E. Boyd, Pamela J. Grizzard, Katherine Hylton Rorie, Santiago Lima

**Affiliations:** 1Department of Biology, Virginia Commonwealth University, Richmond, VA 23284, USA; 2Tissue and Data Acquisition and Analysis Core, Virginia Commonwealth University, Richmond, VA 23298, USA; 3Massey Cancer Center, Richmond, VA 23298, USA

**Keywords:** sphingolipids, lipidomics, ceramide, cancer, health disparities, lung, liver, head and neck, endometrial, colon

## Abstract

**Simple Summary:**

In human cancers, there are molecular changes associated with their development and responses to therapy. These include changes in molecules in their membranes called sphingolipids. Understanding why and how these lipids are altered may lead to new therapies targeting these changes. Unfortunately, as in other areas of modern cancer research, we lack an understanding of how these molecules are altered in tumors of Black Americans, or whether these changes have different impacts than in White Americans. As an important first step, we used mass spectrometry to profile the sphingolipids in tumors of the colon, liver, lung, head and neck, and endometrial cancers of Black and White Americans. We found that sphingolipids with known roles in how tumors grow and respond to therapy are differentially altered in Black and White Americans. These results strongly support further research designed to determine if Black Americans may benefit from therapies that target these alterations.

**Abstract:**

In the US, the incidence and mortality of many cancers are disproportionately higher in African Americans (AA). Yet, AA remain poorly represented in molecular studies investigating the roles that biological factors might play in the development, progression, and outcomes of many cancers. Given that sphingolipids, key components of mammalian cellular membranes, have well-established roles in the etiology of cancer progression, malignancy, and responses to therapy, we conducted a robust mass spectrometry analysis of sphingolipids in normal adjacent uninvolved tissues and tumors of self-identified AA and non-Hispanic White (NHW) males with cancers of the lung, colon, liver, and head and neck and of self-identified AA and NHW females with endometrial cancer. In these cancers, AA have worse outcomes than NHW. The goal of our study was to identify biological candidates to be evaluated in future preclinical studies targeting race-specific alterations in the cancers of AA. We have identified that various sphingolipids are altered in race-specific patterns, but more importantly, the ratios of 24- to 16-carbon fatty acyl chain-length ceramides and glucosylceramides are higher in the tumors of AA. As there is evidence that ceramides with 24-carbon fatty acid chain length promote cellular survival and proliferation, whereas 16-carbon chain length promote apoptosis, these results provide important support for future studies tailored to evaluate the potential roles these differences may play in the outcomes of AA with cancer.

## 1. Introduction

Sphingolipids are one of the major classes of lipids within mammalian membranes [1,2]. In cells, this broad family of metabolites has structural and functional roles and is known to be involved in the regulation of complex biological processes, such as endocytosis, autophagy, cell fate, motility, proliferation, metastasis, angiogenesis, and invasion, among others [3,4,5,6,7,8]. They may influence these by binding to and activating sphingolipid-specific cell surface receptors that trigger intracellular signaling cascades. Glycosylated sphingolipids also form microdomains in membranes that regulate and stabilize cell surface receptors and proteins and can modulate their activity and downstream signaling [9]. Therefore, dysregulation of cellular processes that control sphingolipid levels can significantly impact normal cell function and contribute to the etiology of many diseases, including cancer [8]. In cancers, it is well established that dysregulation of sphingolipid metabolism and of their signaling pathways contributes to disease development, progression, and responses to therapy. Tumors of human patients with various types of cancer have significant alterations of sphingolipid levels as compared to normal adjacent tissues [10,11,12,13,14,15,16,17,18,19,20]. There is also broad consensus that key sphingolipid-metabolism-regulating enzymes impact malignancy, responses to therapy, and prognosis, and regulators of sphingolipid levels are considered important therapeutic targets in cancer, with several being tested in clinical trials [8,21,22,23].

In the US, cancer burden and mortality are disproportionately shouldered by ethnic and racial minorities and individuals from low socioeconomic status. African American (AA) males have historically, and contemporaneously, had significantly higher cancer incidence rates and worse outcomes than non-Hispanic White (NHW) Americans [24]. In the 2022 cancer report to the nation, reporting for the period between 2014 and 2018, for all cancer sites combined, AA males had a 6% higher incidence and 19% higher mortality than NHW males [24]. Compared to NHW, AA males had a higher incidence (per 100k population) for cancers of the prostate (73% higher in AA males), lung (12% higher in AA males), liver (63% higher in AA males), colon (20% higher in AA males), and larynx (41% higher in AA males) [24]. For these cancers, mortality rates were 15% (lung), 130% (prostate), 44% (colon), 56% (liver), and 81% (larynx) higher in AA males compared to NHW males [24]. Even in cancers where AA males had lower incidence, such as those of the oral cavity (46% higher incidence in NHW males), they had higher mortality (7.3% higher in AA males) [24]. There are also disparities between NHW and AA women. For instance, although AA women had a 9% lower incidence for all sites combined, they had a 12% higher mortality rate than NHW women [24]. In breast cancer, AA women had a 3.8% lower incidence, but a 41% higher mortality rate [24]. For cancers of the corpus and uterus, AA and NHW women had similar incidence rates, but AA women had a 96% higher mortality [24]. For cervical cancers, AA women had a 22% higher incidence but 70% higher mortality [24]. AA women have also had higher incidence rates of endometrial cancers since 2002 [25], were more likely to be diagnosed with advanced-stage and aggressive uterine cancers [25], and were less likely to receive guideline-concordant care [26].

It is now understood that the principal underlying causes of health disparities relate to the perpetuation of structural inequalities and systemic discrimination in societal institutions, which significantly decrease social determinants of health among individuals from ethnic and racial minorities, and individuals of low socioeconomic status [27,28]. However, research has also identified geographical-ancestry-related biological differences in cancers that may drive aggressive disease in individuals from admixed populations, impacting outcomes independently of social determinants of health [27]. Therefore, it has been noted that a deeper understanding of all factors, including those impacting social determinants of health and biological variables, which contribute to cancer health disparities, is needed to eradicate these systemic inequalities [29]. Unfortunately, AA and individuals from other ethnic and racial minorities continue to be grossly underrepresented in modern molecular cancer studies and in clinical trials [27,30,31]. Similarly, very little is known about lipidome alterations in the tumors of AA, or whether they play a role in disease outcomes. In addition, given the recognized importance of sphingolipid alterations in the etiology of cancer and their therapeutic potential, precisely establishing how sphingolipid metabolism is altered may lead to the development of targeted therapies. Therefore, to decrease the gap in our understanding of the reprogramming of lipid metabolism in the tumors of individuals self-identifying as AA, we used methodology we previously developed [15,16] to conduct a modern sphingolipidomic analysis of the normal adjacent uninvolved (abbreviated “unin.” henceforth) tissues and tumors of self-identified AA and NHW males with lung adenocarcinoma (LUAD), colorectal adenocarcinoma (COAD), liver adenocarcinoma (HCC), and head and neck squamous cell carcinoma (HNSCC) and of AA and NHW females with endometroid endometrial carcinoma (EEC).

Given that race is a social construct and not a biological variable, geographical genetic ancestry has recently been broadly used to study associations with clinical, epidemiological, or molecular studies. However, as it has been noted, because health disparities are not only the products of genetics and depend on social and environmental factors, reporting findings based on self-identified race remains useful as these may capture associations unlinked from genetic ancestry and influenced by structural inequalities [32]. Therefore, lipidomics data obtained from individuals self-identifying as AA or NHW were analyzed in various ways to gain insight into the reprogramming of sphingolipid metabolism in these individuals, including secondary analyses of self-identified race-dependent and self-identified race-independent changes between unin. tissues and tumors by cancer type, comparisons of the unin. tissues of self-identified AA and NHW by cancer type, and comparison of the tumor tissues of self-identified AA and NHW by cancer type. In addition, to better understand the global reprogramming of sphingolipid metabolism in cancer, a pan-cancer analysis was performed where data for lung, colon, head and neck, and endometrial cancers were combined. Race-dependent and race-independent alterations were evaluated using pan-cancer data. Our study revealed that there are significant differences in the sphingolipid levels and acyl chain-length distribution of unin. tissues and tumors of self-identified AA and NHW.

## 2. Materials and Methods

### 2.1. Specimen Procurement

De-identified human lung tissues cryopreserved in optimal cutting temperature compound (OCT), obtained during standard-of-care procedures and banked when judged to be in excess of that required for patient diagnosis and treatment, were procured from the Virginia Commonwealth University (VCU) Tissue and Data Acquisition and Analysis Core (TDAAC) under a VCU IRB-approved protocol (#HM2471) as described in [15,16]. Samples were provided under an honest broker system, and biorepository participants who signed the TDAAC informed consent documentation agreed to have their residual tissues utilized for any research question, including lipidomic analysis and health information for translational research. Samples were selected based on primary histopathologic confirmed diagnoses and self-identified race. Utilizing institutionally available medical records, age at the time of diagnosis was determined using patient’s date of birth and date of biopsy or definitive surgery. After careful evaluation by a pathologist for histological type, only tumor sections with greater than 35% tumor enrichment and tissues determined to be disease-free (i.e., unin.) were selected for lipidomics studies. There were no significant differences in the tumor enrichment profiles in specimens from AA and NHW for each disease. All associated anonymized sample annotation was generated by the TDAAC SQL2000 secure biorepository database and the Cancer Informatics Core (CIC) VCU-IRB approved protocol (#HM20004765) by way of a dynamic inventory dashboard.

### 2.2. Patient Characteristics

Age, race, and sex characteristics of the patients whose tissues were used in this study are given in Table 1.

### 2.3. Specimen Processing

Prior to mass spectrometry analysis, human specimens were processed with three rounds of the “sOCTrP” method to remove OCT cryoprotectant as described with few modifications [15,16]. Briefly, tissues were resuspended in 10 mL of ice-cold (4 °C) ultrapure Milli-Q water, vortexed for 20 s, centrifuged (5000× *g*, 4 °C) in a swinging bucket centrifuge equipped with ClickSeal biocontainment lids (Thermo Fisher, Whaltam, MA, USA, #75007309), and the supernatant aspirated without disturbing the pellets containing the tissues. After the third round of washing/vortexing/centrifugation, tissues were resuspended in ice-cold phosphate-buffered saline (PBS; Gibco, New York, NY, USA, #20012027), transferred to a preweighed 1.5 mL tube (USA Scientific, Ocala, FL, USA, #1615-5500), centrifuged (7500× *g*, 4 °C), the supernatant removed, and excess liquid removed using a wick made out of laboratory grade tissue paper (Kimtech, Roswell, GA, USA, #34120) (for detailed instructions on wicking please see [15,16]). The tube and tissues were then weighed using an analytical balance and then placed on ice. For accurate data normalization, it was essential to remove as much excess liquid from tissues as possible. Tissue weights were determined as the weight of tubes + tissue minus the weight of the tube alone and used to normalize mass spectrometry data to units of lipid per mg of tissue. Samples that weighed less than 2 mg were excluded from the analysis.

### 2.4. Lipid Extraction

Lipids were extracted as described with minor modifications [15,16]. Briefly, washed and weighed tissues were resuspended in 300 μL of ice-cold (4 °C) PBS, and the tissues transferred to a screw cap 13 × 100 mm screw-top glass tube (VWR, #53283-800) containing 1 mL of LC-MS grade ice-cold methanol and capped (Kimble, Millville, NJ, USA, #45066C-13). Tissues in methanol could be stored at −80 °C until extraction, at which point they were removed from the freezer and allowed to equilibrate to room temperature (23 °C). Then 10 μL of internal standard solution (see below) was added to each tube with a mechanical repeating pipette (Eppendorf, Hauppauge, NY, USA) [15,16]. Tissues were then triturated for 10–20 s in a homogenizer (Homogenizer 150, Fisher Scientific, Waltham, MA, USA), followed by sonication in a Branson 2800 series ultrasonic bath (Branson Ultrasonics, Danbury, CT, USA) for 1–2 min to confirm no tissue clumps remained. If tissue clumps were observed during sonication, the tissues were further processed by repeated cycles of homogenization and sonication. Methanol and chloroform were then added to achieve a 2:1:0.1 ratio of methanol:chloroform:water, and for tissue specimens weighing 50 mg or less, an extraction volume of 3–4 mL was used (volume was adjusted accordingly for larger specimens). Tubes were then tightly capped and incubated overnight at 48 °C. The tubes were then centrifuged (5000× *g*, 10 min, 4 °C), the supernatant decanted into clean 13 × 100 mm borosilicate glass tubes, and the solvent evaporated under vacuum in a Speed-Vac (45 °C). The dried lipids were then resuspended in 600 μL of LC-MS grade methanol by vortexing for 10 s, followed by immersing tubes in a sonicator water bath for 20–30 s. Insoluble debris was then removed by centrifugation at 5000× *g* for 10 min at 4 °C, and the supernatant was carefully decanted into an autoinjector vial (VWR, #46610-724), capped (VWR, #89239-020), and stored at −80 °C until mass spectrometry analysis.

### 2.5. Internal Standards

Internal standards were prepared as described in [15,16]. Briefly, all standards from Avanti Polar Lipids (Alabaster, AL, USA) were added to samples as a cocktail containing 250 pmol each in 10 µL ethanol:methanol:water (7:2:1). Sphingoid base and sphingoid base 1-phosphate standards were 17-carbon chain-length analogs: C17-sphingosine, (2S,3R,4E)-2-aminoheptadec-4-ene-1,3-diol (d17:1-So); C17-sphingosine 1-phosphate, heptadecasphing-4-enine-1-phosphate (d17:1-So1P). Standards for N-acyl sphingolipids were C12-fatty acid analogs: C12-Cer, N-(dodecanoyl)-sphing-4-enine (d18:1/C12:0); C12-Cer 1-phosphate, N-(dodecanoyl)-sphing-4-enine-1-phosphate (d18:1/C12:0-Cer1P); C12-SM, N-(dodecanoyl)-sphing-4-enine-1-phosphocholine (d18:1/C12:0-SM); and C12-glucosylceramide, N-(dodecanoyl)-1-β-glucosyl-sphing-4-eine.

### 2.6. Retention-Time Standards

To assist integration of the correct liquid chromatography electrospray ionization tandem mass spectrometry (LC-ESI-MS/MS) multiple reaction monitoring (MRM) pairs of lipids in tissue samples, a “retention-time standard” sample was prepared by adding 10 µL containing 250 pmol of the following lipids dissolved in ethanol:methanol:water (7:2:1) to 1 mL of LC-MS grade methanol: C16:0-ceramide, N-palmitoyl-D-erythro-sphingosine (d18:1/16:0); C24:1-ceramide, N-nervonoyl-D-erythro-sphingosine (d18:1/24:1(15Z)); C16:0-glucosyl(ß) ceramide, D-glucosyl-ß-1,1′-N-palmitoyl-D-erythro-sphingosine (d18:1/16:0); C24:1-glucosyl(ß) ceramide, D-glucosyl-ß-1,1′-N-nervonoyl-D-erythro-sphingosine (d18:1/24:1(15Z)); 16:0-sphingomyelin (SM), N-palmitoyl-D-erythro-sphingosyl-phosphorylcholine (d18:1/16:0); 24:1-SM, N-nervonoyl-D-erythro-sphingosylphosphorylcholine (d18:1/24:1); C16:0-lactosyl(ß) ceramide, D-lactosyl-ß-1,1′ N-palmitoyl-D-erythro-sphingosine (d18:1/16:0); C24:1-lactosyl(ß) ceramide, D-lactosyl-ß1-1′-N-nervonoyl-D-erythro-sphingosine (d18:1/24:1). All lipids were purchased from Avanti Polar Lipids. Retention-time standards were run separately from samples containing tissue lipid extracts. For example, if 10 tissue samples were analyzed, sample 0 and sample 11 would be retention-time standards. The retention time for each of the lipids in the retention-time standard was noted and used to assist the identification and integration of the correct MRM pairs for their corresponding unknowns in the tissue samples. Because MRM pair retention times shift in a predictable manner that depends on fatty acid carbon chain length and saturation, MRM pair retention times for complex sphingolipids of fatty acid chain lengths 14:0, 18:1, 18:0, 22:0, 24:0, 26:0, and 26:1 could be easily estimated from 16:0 and 24:1 fatty acid chain-length lipids in the retention time standard as previously described [16].

### 2.7. Mass Spectrometry Analysis

Analysis was performed as described in [15,16,33]. For LC-ESI-MS/MS analyses, a Shimadzu Nexera LC-30 AD binary pump system coupled to a SIL-30AC autoinjector and DGU20A5R degasser coupled to an AB Sciex 5500 quadrupole/linear ion trap (QTrap) (SCIEX, Framingham, MA, USA) operating in a triple quadrupole mode were used. Q1 and Q3 were set to pass molecularly distinctive precursor and product ions (or a scan across multiple *m*/*z* in Q1 or Q3), using N2 to collisionally induce dissociations in Q2 (which was offset from Q1 by 30–120 eV); the ion source temperature set to 500 °C. For Q1 and Q3 precursor and product ion transitions, please see refs. [14] and [15], respectively. Sphingolipids were separated by reverse-phase LC on a Supelco 2.1 × 50 mm Ascentis Express C18 column (Sigma, St. Louis, MO, USA) at 35 °C using a binary solvent system (flow rate of 0.5 mL/min). Prior to injection, the column was equilibrated for 0.5 min with a solvent mixture of 95% mobile phase A1 (CH3OH:water:HCOOH, 58:41:1, *v*:*v*:*v*, with 5 mM ammonium formate) and 5% mobile phase B1 (CH3OH:HCOOH, 99:1, *v*:*v*, with 5 mM ammonium formate), and after sample injection (typically 5 μL), the A1/B1 ratio was maintained at 95/5 for 2.25 min, followed by a linear gradient to 100% B1 over 1.5 min, which was held at 100% B1 for 5.5 min, followed by a 0.5 min gradient return to 95/5 A1/B1. The column was re-equilibrated with 95/5 A1/B1 for 0.5 min before the next run.

### 2.8. Statistical Analysis

All statistical analyses were performed using Prism 8 (GraphPad, Boston, MA, USA). Mass spectrometry MRM-pair-integrated peak areas normalized per mg of tissue for analyzed lipids were examined for normality and lognormality using Anderson–Darling and D’Agostino–Pearson tests. Normalized mass spectrometry data were grouped by primary histopathologic confirmed diagnoses and self-identified race. Descriptive statistics and differences in patient age at the time of surgery were performed using unpaired two-tailed *t*-tests and α = 0.05. For self-identified race-independent analysis of changes between unin. tissues and tumors, and self-identified race-dependent differences between unin. tissues and tumors, data were examined by ANOVA followed by Fisher’s least significant difference (LSD) test with a 95% confidence interval for family-wise significance and confidence level. Given our intent to find all potential statistically significant alterations, uncorrected *p*-values were used to determine significance (α = 0.05) and are shown for all figures. However, because of this approach, for analysis that evaluates differences between unin. and tumor tissues of AA and NHW, the 95% family-wise significance and confidence intervals for each comparison were plotted and shown in Section 3.3, Section 3.4 and Section 3.5. In addition, *p*-adjusted values were also calculated after corrections for multiple comparisons following ANOVA by controlling for the false discovery rate (Q = 0.05) and the two-stage step-up method of Benjamini, Krieger, and Yekutieli (Prism). Multiple comparison-adjusted *p*-values (q) are given in Appendix A alongside calculated uncorrected LSD *p*-values.

## 3. Results

### 3.1. Self-Identified Race-Independent and Disease-Specific Analysis of Sphingolipid Metabolism Alterations

Tissues cryopreserved in optimal cutting temperature compound were procured from a biorepository, processed, and LC-ESI-MS/MS as described in [15,16]. When data from NHW and AA were combined, relative to unin. tissues, the tumors of male subjects diagnosed with LUAD had significantly higher levels of ceramides of 16:0, 24:0, and 24:1 acyl chain lengths (Figure 1A); 16:0, 24:0, and 24:1 monohexosylceramide (monoHexCer; Figure 1B); and 16:0 lactosylceramide (lacCer; Figure 1D). In LUAD, there were no significant changes in any acyl chain length of SM (Figure 1C), nor in any of the sphingoid bases (Figure 1U). In EEC tumors from AA and NHW females combined, compared to unin. tissues, there were significantly higher levels of 16:0, 22:0, 24:1, and 24:0 ceramide (Figure 1E); 16:0, 22:0, 24:1, and 24:0 monoHexCer (Figure 1F); 14:0, 16:0, 22:0, 24:1, and 24:0 SM (Figure 1G); and 16:0, 24:1, and 24:0 lacCer (Figure 1H). There were also significantly higher levels of 18:1 sphingosine (So) and 18:0 dihydrosphingosine (Sa; Figure 1V), but not of So-1-phosphate (So-1-P) or Sa-1-phosphate (Sa-1-P; Figure 1V). COAD tumors from AA and NHW males combined had significantly higher levels of 16:0, 24:1, and 24:0 ceramide (Figure 1I); 16:0, 22:0, 24:1, and 24:0 monoHexCer (Figure 1J); 14:0 and 16:0 SM (Figure 1K); 16:0, 24:1, and 24:0 lacCer (Figure 1L), So and Sa (Figure 1W). COAD tissues also had significantly lower levels of 18:0 and 20:0 SM (Figure 1K) and no changes in So-1-P or no Sa-1-P (Figure 1W). HCC tumors from AA and NHW males combined had no significant changes in any species of ceramide (Figure 1M) or SM (Figure 1O), but significantly lower 22:0 and 24:0 monoHexCer (Figure 1N), and lower 16:0 and 22:0 lacCer (Figure 1P). HCC tumors also had significantly lower So, Sa, and So-1-P, but no changes in Sa-1-P (Figure 1X). In HNSCC tumors from male AA and NHW combined, there were significantly higher levels of 16:0, 22:0, 24:1, 24:0, and 26:0 ceramides (Figure 1Q); 16:0 and 24:0 monoHexCer (Figure 1R); 26:1 SM (Figure 1S); and 16:0, 24:1, and 24:0 lacCer (Figure 1T), but no significant differences in the levels of So, Sa, So-1-P, and Sa-1-P (Figure 1Y). In Figure 1, only unadjusted *p*-values are shown. For FDR/BKY-adjusted *p*-values (Q = 5%) and means differences (means diff.), please see Appendix A.

### 3.2. Disease-Specific Analysis of Sphingolipid Metabolism Alterations in Self-Identified AA Males and Females

In AA males diagnosed with LUAD, relative to unin. tissues, tumors had significantly higher 24:1 ceramide (Figure 2A); 16:0, 24:1, and 24:0 monoHexCer (Figure 2B); 16:0 lacCer (Figure 2D); and So (Figure 2U), but no significant changes in SM of any acyl chain length (Figure 2C) or of Sa or the phosphorylated sphingoid bases (Figure 2U). In AA females diagnosed with EEC, tumors had significantly higher levels of 22:0, 24:1, and 24:0 ceramide (Figure 2E); 16:0, 22:0, 24:1, and 24:0 monoHexCer (Figure 2F); 14:0, 16:0, 22:0, 24:1, and 24:0 SM (Figure 2G); 16:0 and 24:0 lacCer (Figure 2H); and So and Sa (Figure 2V), but not of So-1-P or Sa-1-P (Figure 2V). In AA males with COAD, tumors had significantly higher levels of 16:0, 24:1, and 24:0 ceramide (Figure 2I); 16:0, 24:1, and 24:0 monoHexCer (Figure 2J); 14:0 and 16:0 SM (Figure 2K); 16:0 and 24:1 lacCer (Figure 2L); and So and Sa, but not So-1-P or Sa-1-P (Figure 2W). Tumor tissues of AA males with COAD also had significantly lower levels of 18:0 and 20:0 SM (Figure 2K). In AA males diagnosed with HCC, tumors had significantly lower 24:0 monoHexCer (Figure 2N) and 16:0 lacCer (Figure 2P) and Sa and So-1-P (Figure 2X). There were no significant changes in the tumors of AA males with HCC in any chain length of ceramide (Figure 2M) or SM (Figure 2O), or of So or Sa-1-P (Figure 2X). The tumors of AA males diagnosed with HNSCC had significantly higher 16:0, 22:0, 24:1, 24:0, and 26:0 ceramide (Figure 2Q); 16:0, 24:1, and 24:0 monoHexCer (Figure 2R); and 16:0, 24:1, and 24:0 lacCer (Figure 2T), but no changes in SM (Figure 2S) or any of the sphingoid bases (Figure 2Y). In Figure 2, unadjusted *p*-values are shown. For FDR/BKY-adjusted *p*-values (Q = 5%) and means differences (means diff.), please see Appendix A.

### 3.3. Disease-Specific Analysis of Sphingolipid Metabolism Alterations in Self-Identified NHW Males and Females

In NHW males diagnosed with LUAD, tumors had significantly higher 16:0, 24:1, and 24:0 ceramide (Figure 3A); 16:0, 24:1, and 24:0 monoHexCer (Figure 3B); and 16:0 lacCer (Figure 3D), but no significant changes in SM of any acyl chain length (Figure 3C) or any of the sphingoid bases (Figure 3U). In NHW females diagnosed with EEC, tumors had significantly higher levels of 16:0, 22:0, 24:1, and 24:0 ceramide (Figure 3E); 16:0, 22:0, 24:1, and 24:0 monoHexCer (Figure 3F); 22:0, 24:1, and 24:0 SM (Figure 3G); 16:0, 24:1, and 24:0 lacCer (Figure 3H); and So and Sa (Figure 3V), but not of So-1-P or Sa-1-P (Figure 3V). In NHW males diagnosed with COAD, tumors had significantly higher levels of 16:0, 24:1 and 24:0 ceramide (Figure 3I); 16:0, 24:1, and 24:0 monoHexCer (Figure 3J); 16:0 and 24:1 lacCer (Figure 3L); and So and Sa (Figure 3W). COAD NHW tumor tissues had significantly lower 18:0 and 20:0 SM (Figure 3K), but no changes in So-1-P and Sa-1-P (Figure 3W). Tumors of NHW males diagnosed with HCC had significantly higher 24:1 ceramide (Figure 3M), but no significant differences in SM (Figure 3O), lacCer (Figure 3P), or Sa, So-1-P, and Sa-1-P (Figure 3X). However, HCC tumors from NHW had significantly lower 22:0 and 24:0 monoHexCer (Figure 3N) and So (Figure 3X). In NHW males diagnosed with HNSCC, tumors had significantly higher 16:0, 24:1, 24:0, and 26:0 ceramide (Figure 3Q) and 16:0, 24:1, and 24:0 lacCer (Figure 3T), but no significant changes were observed in any acyl chain length of monoHexCer (Figure 2R), SM (Figure 3S) or any of the sphingoid bases (Figure 3Y).

### 3.4. Disease-Specific Comparisons between Self-Identified AA and NHW with Cancer

Sphingolipid alterations have been associated with the etiology of cancer progression, malignant phenotypes, and responses to therapy. Therefore, a critical objective of this study was to explore differences in the sphingolipid tumor biology of self-identified AA who may be promising actionable targets to evaluate in preclinical cancer research models. To reveal potential differences, the following comparisons were made for each cancer analyzed: AA unin. • NHW unin. and AA tumor • NHW tumor. The tumors of self-identified AA males with LUAD, compared to self-identified NHW males, had significantly lower levels of 24:1 ceramide (Figure 4A), 16:0 and 24:1 lacCer (Figure 4D), and So and Sa (Figure 4E). However, the unin. tissues of AA males with LUAD had higher levels of 18:0 and 22:0 SM (Figure 4C). When comparing the tissues of AA females with EEC to NHW females with EEC, we observed that the tumors of AA females had significantly lower levels of 24:1 ceramide (Figure 5A), 16:0 monoHexCer (Figure 5B), 24:1 SM (Figure 5C), and 16:0 lacCer (Figure 5D), but significantly higher levels of So (Figure 5E). No differences were observed in AA unin. and NHW unin. comparisons between AA and NHW females with EEC (Figure 5A–E). Between AA and NHW male subjects with COAD, the tumors of AA males with COAD had significantly higher levels of 24:1 ceramide (Figure 6A), 16:0 SM (Figure 6C), and 16:0 lacCer (Figure 6D), but lower levels of Sa (Figure 6E). The unin. tissues of AA males with COAD had higher levels of 24:1 ceramide (Figure 6A) and 24:0 monoHexCer (Figure 6B). In males with HCC, the tumor tissues of AA males, compared to NHW males, had significantly higher 24:1 and 24:0 ceramide (Figure 7A), and the unin. tissues of AA males had significantly higher levels of 24:1 ceramide (Figure 7A), 24:0 SM (Figure 7C), 16:0 lacCer (Figure 7D), and So and Sa (Figure 7E), but lower levels of 22:0 monoHexCer (Figure 7B). In HNSCC, the tumors of AA males had significantly higher 16:0 and 24:0 ceramide (Figure 8A) and 16:0, 22:0, 24:1, and 24:0 monoHexCer (Figure 8B). The unin. tissues of AA males with HNSCC had significantly lower 16:0 lacCer (Figure 8D).

### 3.5. Pan-Cancer Analysis of Sphingolipid Metabolism Reprogramming

The term cancer describes a large group of related diseases broadly encompassing molecularly and histopathologically diverse forms now understood to originate from distinct somatic driver mutations and clonal selection pathways. Advances in our understanding of driver-specific origins of each disease, along with molecular histopathological typing, have revolutionized disease-specific therapy and improved outcomes. Yet, cancers of different histopathological and genetic origins still share many phenotypic, genotypic, and metabolic properties. Hence, our molecular understanding of cancer includes well-accepted hallmarks that describe traits positively selected during carcinogenesis and transformation that are common to all disease forms, for instance, reprogramming of energy-producing pathways in transformed cells, known as the Warburg effect, and changes in the balance of lipid anabolism and catabolism in non-lipid-producing cells, better known as lipogenesis. Therefore, a comprehensive understanding of the commonalities and shared molecular patterns across many different types of cancer can lead to a better understanding of their etiology and to the development of actionable therapeutic targets broadly applicable to several diseases. This “pan-cancer” approach with pooled datasets from cancers of many different histologies has been used to better understand the overarching themes in genomic and driver-led architectures [34,35,36]. Thus, to gain a better understanding of the global patterns of sphingolipid metabolism reprogramming in human cancers, we conducted two secondary “pan-cancer” analyses: (a) by combining data from AA and NHW with LUAD, EEC, COAD, HCC, and HNSCC and (b) by combining data from all cancers, but grouping the data by race.

In the race-independent pan-cancer analysis where data from AA and NHW were combined, pan-cancer tumors had significantly increased ceramides of 16:0, 22:0, 24:1, 24:0, and 26:0 chain length (Figure 9A); monoHexCer of 16:0, 24:1, and 24:0 chain length (Figure 9B); and lacCer of 16:0, 24:1, and 24:0 chain length (Figure 9D). So and Sa were also significantly higher in pan-cancer tumors (Figure 9E). There were no significant changes observed in any chain length of SM (Figure 9C), So-1-P or Sa-1-P (Figure 9E). In the self-identified race-dependent pan-cancer analysis that combined all cancer histology data but kept data from AA and NHW separated, the following comparisons were made: AA unin. • NHW unin., AA tumor • NHW tumor, which are shown in Figure 10. The race-dependent pan-cancer analyses revealed that the tumors of AA, relative to those of NHW, were significantly enriched in 24:1 and 24:0 acyl chain-length ceramides (Figure 10A). Tumors of AA also had significantly higher 16:0 and 24:0 monoHexCer (Figure 10B), 16:0 lacCer (Figure 10D), and So (Figure 10E). No changes in any acyl chain length of SM were noted in either unin. or tumor tissues (Figure 10C). However, the unin. tissues of AA were observed to be significantly enriched with 24:1 ceramide (Figure 10A).

## 4. Discussion

In the US, cancer health disparities are rooted in systemic racism and institutional inequalities that limit access to health care, early detection services, health insurance, health literacy, nutritious food, and education and are influenced by socioeconomic status and a lack of diversity among healthcare providers [27,28]. Significant inequalities are also prevalent in the delivery of standard-of-care therapy to individuals with cancer from racial and ethnic minorities and of those from low socioeconomic status, along with a poor representation of minorities in clinical trials and modern molecular cancer studies [27,28]. These structural and social inequalities detrimentally impact social determinants of health and decrease the quality of life of individuals from racial and ethnic minorities and of those of low socioeconomic status.

Increased social awareness and accountability within the basic sciences and healthcare industry, along with a concomitant expansion in cancer health disparities funding by the National Institutes of Health, has led to an increase in research geared to better understand the complex roots of cancer disparities, including those of biological origin. In particular, the past decade has seen the discovery of differences in the genomic and transcriptomic signatures of tumors from AA and NHW. These include genome-wide variances in recombination repair deficiency [37,38], chromothripsis [37], genomic instability [37], RNA splicing [39], DNA “repairome” genes [40], and race-specific driver mutations associated with poor outcomes [41,42,43,44,45]. As recognition of the importance of understanding cancer metabolism reprogramming has grown [46,47], research into race-specific metabolism changes in cancer has also grown with it. This has led to the identification of differences in how nicotine and cigarette smoking carcinogens are metabolized in AA [48,49,50,51,52,53,54] (for review see [55]), metabolomic biomarkers of aggressive disease [56,57,58,59,60], lipidomic signatures in the plasma and serum of AA and NHW with cancer [57,58,59,61,62], fatty acid analysis in prostate cancer [63], and lipidomic analysis of tumors of AA women with breast cancer [64]. Yet, as in many other areas of modern molecular cancer research and clinical trials, AA remain largely underrepresented in modern tumor lipidomic analysis studies. In this study, LC-ESI-MS/MS was used to analyze the normal adjacent uninvolved tissues and tumor tissues of AA and NHW diagnosed with LUAD, EEC, COAD, HCC, and HNSCC. Secondary analyses were conducted with these data separated by disease (Figure 1) and by self-identified race and disease (Figure 2, Figure 3, Figure 4, Figure 5, Figure 6, Figure 7 and Figure 8). In addition, data were combined by disease and analyzed in a self-identified race-dependent (Figure 9) and -independent manner (Figure 10).

Significant increases in ceramides, monoHexCer, and lacCer of various fatty acid chain lengths were observed in the tumors of AA and NHW with LUAD, EEC, COAD, and HNSCC. Interestingly, the sphingolipid alterations observed in HCC were dramatically different from the other cancers analyzed. Although in the other cancers most sphingolipids were increased, in HCC most significant alterations between unin. tissues and tumors were decreased. SM alterations were only observed in the tumors of individuals from both races with COAD and EEC. Interestingly, however, significant COAD SM alterations were both positive and negative. However, in both race-dependent and race-independent pan-cancer analyses, no SM of any chain length was significantly altered. Pan-cancer analyses also revealed a significant increase in ceramides, monoHexCer, and lacCer of 16:0, 24:1, and 24:0 chain lengths, So and Sa, as well as ceramides of 22:0 and 26:1 fatty acyl chain length.

Ceramides of 24- or 16-carbon fatty acyl chain length are known to differentially impact cell fate (for reviews see [65]). Evidence suggests that ceramides with 24-carbon fatty acid acyl chain lengths are pro-survival lipids as they have cytoprotective effects, promote cellular growth, and have anti-apoptotic properties [66,67,68]. However, ceramides with 16-carbon fatty acid acyl chain length can induce cellular stress [66,69], have pro-apoptotic properties that decrease cell growth [67,68,70,71,72], can induce mitochondrial outer membrane permeabilization [73], and induce apoptosis by specifically binding to and activating the tumor-suppressor p53 to induce its downstream pro-apoptotic targets and modulating cell growth [74]. For mitochondrial outer membrane permeabilization, the ratio of 24-carbon to 16-carbon fatty acid chain-length ceramides (referred to as “24•16 ceramide ratio” henceforth) influences the capacity of ceramide to induce apoptosis, with higher ratios (i.e., higher levels of 24-carbon ceramide) inhibiting the 16-carbon length ceramide to induce mitochondrial outer membrane permeabilization [72,75]. Furthermore, the 24•16 ceramide ratio influences cell fate in cancer cells treated with ionizing radiation, with higher ratios (higher 24-carbon ceramide) protecting cells and lower ratios (higher 16-carbon length ceramides) promoting apoptosis [68]. In colon cancer cells, the 24•16 ceramide ratio can also significantly impact cell fate by promoting cellular proliferation (with higher 24-carbon ceramide) or inducing apoptosis (with higher 16-carbon ceramide) [67].

Except for tumors from AA males with HCC, 24:1 ceramide was significantly increased in tumors of all cancers analyzed, and 16:0 ceramides were also significantly elevated in most tumors, including those of NHW with LUAD, EEC, COAD, and HNSCC, and in the tumors of AA with COAD and HNSCC. However, we found that 16:0 ceramides were not significantly elevated in the tumors of AA males with LUAD or AA females with EEC (Figure 2A,E). This led us to consider the possibility that tumors from AA may have higher 24•16 ceramide ratios than NHW and suggest potentially important tumor biology differences. The 24•16 tumor ceramide ratios were examined as follows: Using mean normalized tumor values, three different ratios were estimated for each race: (a) a “24:1/24:0 • 16” ratio, where the mean tumor 24:1 and 24:0 ceramides values were added together and then divided by the mean tumor 16:0 ceramide value; (b) a “24:1•16:0” ratio, where the mean tumor 24:1 ceramide value was divided by the mean tumor 16:0 ceramide value; and (c) a “24:0•16:0” ratio, where the mean tumor 24:0 ceramide value was divided by the mean tumor 16:0 ceramide value. Larger ratios suggest that the relative levels of 24-carbon ceramide are higher than 16-carbon ceramide. In LUAD tumors, the 24:1/24:0 • 16 ratio was 26% higher in AA, the 24:1•16:0 was 20% higher in AA, and the 24:0•16:0 was 32% higher in AA. In EEC tumors, the 24:1/24:0 • 16 ratio was 29% higher in AA, the 24:1•16:0 was 1.2% higher in AA, and the 24:0•16:0 was 56% higher in AA. In COAD tumors, the 24:1/24:0 • 16 was 25% higher in AA, the 24:1•16:0 was 38% higher in AA, and the 24:0•16:0 was 3.1% higher in AA. In HCC tumors, the 24:1/24:0 • 16 ratio was 89% higher in AA, the 24:1•16:0 was 75% higher in AA, and the 24:0•16:0 was 102% higher in AA. In HNSCC tumors, the 24:1/24:0 • 16 ratio was 7% lower in AA, the 24:1•16:0 was 16% lower in AA, and the 24:0•16:0 was 1% higher in AA. For pan-cancers, the 24:1/24:0 • 16 ratio was 16% higher in AA, the 24:1•16:0 was 19% higher in AA, and the 24:0•16:0 was 13% higher in AA. Importantly, calculations based on a geometric mean or median 24:1, 24:0, and 16:0 tumor ceramide values produced similar results, with AA having higher ratios in LUAD, EEC, COAD, and HCC, but not in HNSCC.

There are two important caveats to these observations: (1) In LUAD and EEC, when the tissues of AA and NHW were compared (Figure 4 and Figure 5), although mean levels of 16:0 ceramide in unin. and tumor tissues were lower in AA, these differences did not reach significance. (2) The tumor levels of 24:1 ceramide (but not 24:0 ceramide) were significantly lower in AA males with LUAD and AA females with EEC (Figure 4A and Figure 5A). However, these concerns may be diminished by the observation that the magnitude of the change in 16:0 ceramides between unin. tissues and tumors was 54% larger in NHW males with LUAD and 52% larger in NHW females with EEC. These changes are reflected in observations discussed above, as the levels of 16:0 ceramide were significantly increased in the LUAD and EEC tumors of NHW, but they did not change in tumors of AA with the same disease. These caveats are less of a concern in COAD and HCC, as differences in 24•16 ceramide ratios can be more easily explained by the significantly higher tumor levels of 24:1 and 24:0 ceramides (Figure 6A and Figure 7A). Consistent with this logic, in HNSCC, where the 24•16 ceramides ratios were similar between AA and NHW, the tumors of AA had significantly higher 16:0 and 24:0 ceramide than NHW (Figure 8A).

In combination, we speculate these results suggest that differences in 24•16 ceramide ratios may be due to higher overall relative levels of the 24:1 and 24:0 fatty acid chain-length ceramides in the tumors of AA. Indeed, in the pan-cancer analysis, globally, the tumors of AA had significantly higher levels of 24:1 and 24:0 ceramide (Figure 10A), but the levels of 16:0 ceramide in unin. tissues or tumors were not significantly different (Figure 10A). As discussed above, given that the 24•16 ceramide ratio can tilt the cellular pro-apoptotic/pro-survival balance, one interpretation of these results is that tumors of AA have ceramide profiles indicative of a more pro-survival phenotype. However, it is important to note that given the modest number of samples used for each disease, further studies with larger cohorts will be needed to confirm these observations. Moreover, it will be critical to assess if there are associations between self-identified race, geographical ancestry, and 24•16 ceramide ratios with poor outcomes, responses to therapy, and malignancy in these or other cancers.

Aside from critically important functions in the plasma membrane, monoHexCers, like the glycosphingolipid glucosylceramide, are strongly tied to the pathophysiology of acquired multi-drug chemoresistance in tumor-derived cell lines and in the tumors of patients who have failed chemotherapy (for reviews see [76,77]). In cancer cells, elevated levels of glucosylceramide are intimately associated with the increased expression of multi-drug resistance transporter proteins that act as drug efflux pumps that decrease their cytotoxic efficacy. Therefore, cataloging monoHexCer alterations in human tumors may lead to a better understanding of molecular mechanisms driving chemoresistance in patients. In our race-dependent analysis, with the exception of tumors from AA and NHW with HCC and NHW males with HNSCC, monoHexCer of 16:0, 24:1, and 24:0 acyl chain length was significantly increased in the tumors of AA and NHW with LUAD, EEC, and COAD and in AA males with HNSCC. Intriguingly, although the tumors of AA males with HNSCC had significantly elevated 16:0, 24:1, and 24:0 monoHexCer (Figure 2R), the tumors of NHW males had no significant changes in monoHexCer of any chain length (Figure 3R). As there are well-documented disparities in the outcomes of laryngeal and oral squamous cell carcinomas among AA and NHW males, it is interesting to consider if outcomes in this disease are influenced by changes in monoHexCer (as they are higher in AA males) rather than the 24•16 ceramide ratio as discussed above (which was lower in AA males with HNSCC). Interestingly, in clinical trials, where AA participants were study-arm-matched with NHW controls, thereby minimizing socioeconomic and access to healthcare confounders, AA still had a 51% significantly higher risk of locoregional failure [78]. Therefore, it may be important to investigate if there is an association between AA males with HNSCC who have failed chemotherapy and remodeling of their 24•16 ceramide ratios relative to changes in monoHexCer profiles. Importantly, the pan-cancer race-dependent analysis also revealed that globally the tumors of AA had significantly higher levels of 16:0 and 24:0 monoHexCer (Figure 10B).

Little is known about the role of lacCer alterations in human cancers. However, it has been shown that overexpression of the enzyme that synthesizes it increases the proliferation of cancer cells [79], and its levels are significantly elevated in colorectal cancer [20]. In addition, the enzyme of the B3GNT5 gene product, which converts lacCer to lactotriaosylceramide, has been found to be significantly elevated in the tumors of patients with glioblastoma multiforme and was associated with decreased overall survival [80]. Moreover, a long noncoding RNA that increases the expression of B3GNT5 was shown to be positively associated with poor prognosis and metastasis in patients with liver cancer [81]. In this report, lacCer of various chain lengths was significantly elevated in the tumors of AA and NHW patients with LUAD, EEC, COAD, and HNSCC (Figure 1). LacCer of various chain lengths was also significantly lower in tumors of AA males with LUAD (Figure 4D) and females with EEC (Figure 5D), but higher in the tumors of AA males with COAD (Figure 6D), HCC (Figure 7D), and HNSCC (q = 9.0 × 10^−3^, Appendix A). However, it is noted that in the pan-cancer analysis, lacCer of 16:0 chain length was significantly higher in the tumors of AA (Figure 10D). Unfortunately, given how little is known about the role of lacCer in the etiology of cancer progression and malignancy, and the mixed effects we see in reprogramming of this lipid by disease type and race, it is difficult to speculate on the potential implications of these differences without conducting further detailed molecular studies.

Our study is a vital first characterization of sphingolipidome alterations in the tumors of AA with these cancers. However, given our results, further confirmation using a larger cohort of AA and NHW for each disease is warranted. Importantly, we should point out that procuring specimens adequate for mass spectrometry can be challenging as formalin-fixed tissues are not suitable for this type of analysis [16]. Moreover, procuring tissues amenable for mass spectrometry from AA males is extremely challenging, as the lack of representation of these individuals in modern studies and clinical trials is reflected by their poor representation in biorepositories [29]. Therefore, even with modest subjects per group and disease, our study significantly advances our understanding of the tumor biology of AA and reveals some potential actionable differences to explore using preclinical models and studies exploring associations and correlations between clinicopathological variables and lipidomics data.

We previously described methods for the preparation of human tissues cryopreserved in OCT for mass spectrometry analysis [16]. In rigorously controlled experiments, results indicated that three tissue washes (as performed here) were sufficient to remove OCT and accurately and without any significant losses determine the levels of ceramides, monoHexCer, SM, lacCer, So, and Sa [16]. However, these experiments also revealed potential losses of tissue-associated So-1-P and Sa-1-P, and it was suggested that the interpretation of their levels using this method should be carried out with caution [16]. Therefore, in this study, findings showing minor changes in So-1-P and Sa-1-P should be carefully evaluated within the context of this technical limitation, especially in light of the many roles that have been ascribed to So-1-P and associated signaling pathways in cancer [8]. To overcome this limitation, we suggest that further characterization of tissues not cryopreserved in OCT may be required to accurately assess whether there are differences in the levels of So-1-P and Sa-1-P in the tissues of AA and NHW with cancer.

In this study, self-identified race was used to partition subjects in race-dependent analyses. Therefore, we recognize that our findings are not generalizable to studies evaluating associations between sphingolipidome alterations in tumors and geographical admixture genetic ancestry. However, even in the context of these caveats and limitations, there are several findings of potentially significant biological impact, and follow-up studies may lead to insights into mechanisms contributing to aggressive disease in AA when socioeconomic variables are carefully controlled for.

## 5. Conclusions

In this report, we have expanded the molecular characterization of tumors from self-identified AA with various types of cancer and made comparisons with similar tissues from NHW. Significant variations in the sphingolipid profiles of tumors and unin. tissues were observed, which suggest some potentially significant molecular differences in the biology of AA and NHW tumors. Therefore, we suggest that these results warrant further analysis with larger patient cohorts and in vitro studies that evaluate the impact that heterogeneity in sphingolipid metabolism has on the phenotypic properties of transformed cells from each race. Future studies could establish potential actionable molecular targets that specifically decrease tumor malignancy and improve outcomes in AA with cancer.

## Figures and Tables

**Figure 1 cancers-15-02238-f001:**
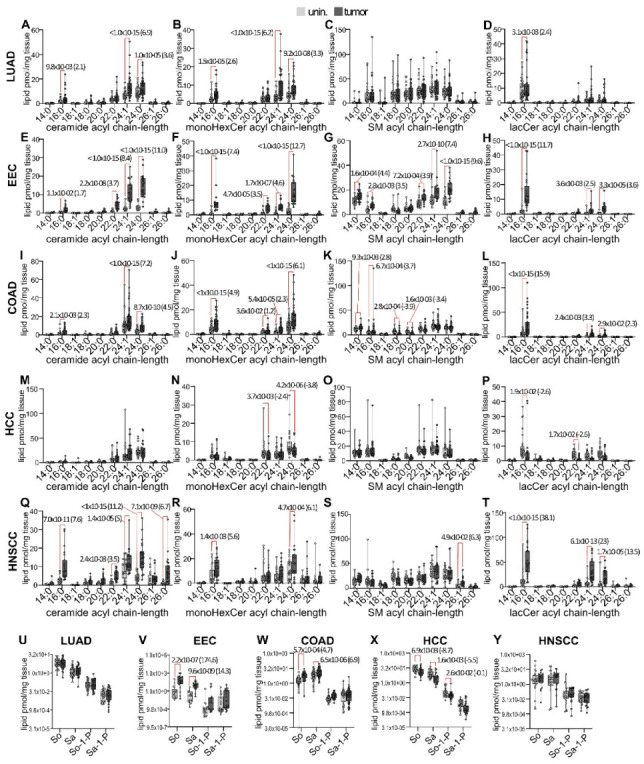
Sphingolipid profiles of normal adjacent uninvolved tissues and tumors of male patients diagnosed with LUAD, COAD, HCC, and HNSCC and of female patients with EEC. The indicated chain-length sphingolipids were quantified by LC-ESI-MS/MS in normal adjacent uninvolved tissues (unin.) and tumors of self-identified AA and NHW subjects (data combined in plots) with primary histopathologic confirmed diagnoses of lung adenocarcinoma (LUAD; panels (**A**–**D**,**U**)), endometrial endometroid carcinoma (EEC; panels (**E**–**H**,**V**)), colorectal adenocarcinoma (COAD; panels (**I**–**L**,**W**)), hepatocellular adenocarcinoma (HCC; panels (**M**–**P**,**X**)), and head and neck squamous cell carcinoma (HNSCC; panels (**Q**–**T**,**Y**)). Lipid levels are shown as the amount of picomoles of indicated chain length and lipid species per mg of tissue. For LUAD, unin. n = 34, tumor n = 40; EEC, n = 22; COAD, n = 54; HCC, n = 21; HNSCC, unin. n = 24, tumor n = 23. Graphs are box plots showing medians (black line) and whiskers of min to max. Uncorrected *p*-values for associations considered significant (α ≤ 0.05) are shown, with values of the predicted mean difference (ANOVA) between unin. and tumor tissue lipid levels shown in parenthesis. The lower limit of *p*- and q-value calculations was set to 1.0 × 10^−15^ (Prism), so no values are reported lower than this limit. For p-adjusted (q) values corrected for multiple comparisons using the false discovery rate (FDR; Q = 0.05) and the two-step controlling procedure of Benjamini, Krieger, and Yekutieli (BKY), see Appendix A. For panels (**U**–**Y**), the *y*-axis is in log2 scale.

**Figure 2 cancers-15-02238-f002:**
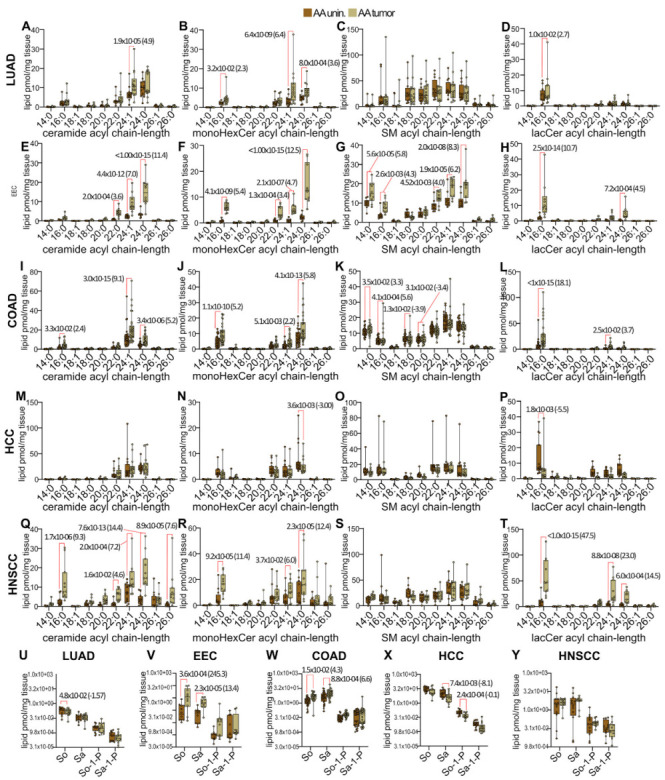
Sphingolipid profiles of normal adjacent uninvolved tissues and tumors of self-identified AA males diagnosed with LUAD, COAD, HCC, and HNSCC and of self-identified AA females with EEC. The indicated chain-length sphingolipids were quantified by LC-ESI-MS/MS in normal adjacent uninvolved tissues (unin.) and tumors of self-identified AA subjects with primary histopathologic confirmed diagnoses of lung adenocarcinoma (LUAD; panels (**A**–**D**,**U**)), endometrial endometroid carcinoma (EEC; panels (**E**–**H**,**V**), colorectal adenocarcinoma (COAD; panels (**I**–**L**,**W**)), hepatocellular adenocarcinoma (HCC; panels (**M**–**P**,**X**)), and head and neck squamous cell carcinoma (HNSCC; panels (**Q**–**T**,**Y**)). Lipid levels are shown as the amount of picomoles of indicated chain length and lipid species per mg of tissue. For LUAD, unin. n = 12, tumor n = 14; EEC, n = 10; COAD, n = 30; HCC, n = 10; HNSCC, unin. n = 12, tumor n = 11. Graphs are box plots showing medians (black line) and whiskers of min to max. Uncorrected *p*-values for associations considered significant (α ≤ 0.05) are shown, with values of the predicted mean difference (ANOVA) between unin. and tumor tissue lipid levels shown in parenthesis. The lower limit of *p*- and q-value calculations was set to 1.0 × 10^−15^ (Prism), so no values are reported lower than this limit. For *p*-adjusted (q) values corrected for multiple comparisons using the FDR (Q = 0.05) and the two-step controlling procedure of BKY, see Appendix A. For panels (**U**–**Y**), the *y*-axis is in log2 scale.

**Figure 3 cancers-15-02238-f003:**
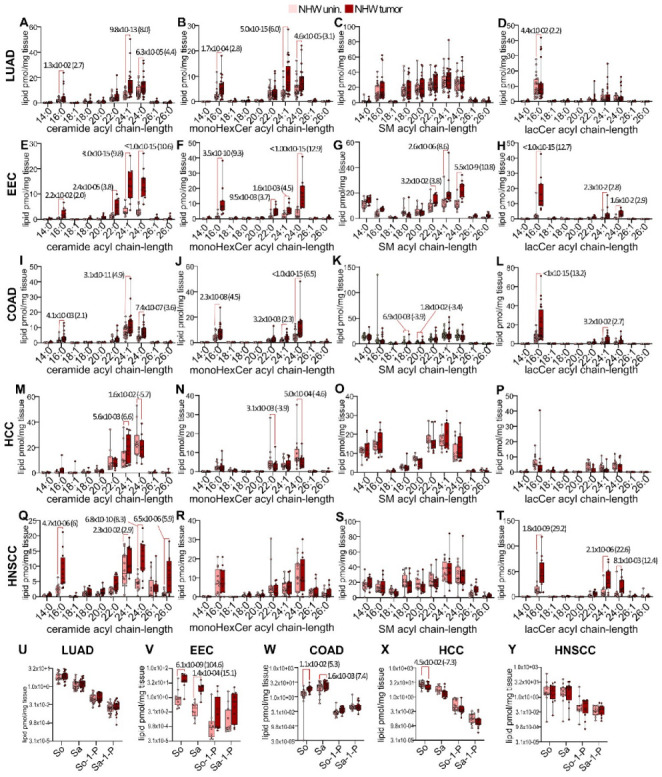
Sphingolipid profiles of normal adjacent uninvolved tissues and tumors of self-identified NHW males diagnosed with LUAD, COAD, HCC, and HNSCC and of self-identified NHW females with EEC. The indicated chain-length sphingolipids were quantified by LC-ESI-MS/MS in normal adjacent uninvolved tissues (unin.) and tumors of self-identified NHW subjects with primary histopathologic confirmed diagnoses of lung adenocarcinoma (LUAD; panels (**A**–**D**,**U**)), endometrial endometroid carcinoma (EEC; panels (**E**–**H**,**V**)), colorectal adenocarcinoma (COAD; panels (**I**–**L**,**W**)), hepatocellular adenocarcinoma (HCC; panels (**M**–**P**,**X**)), and head and neck squamous cell carcinoma (HNSCC; panels (**Q**–**T**,**Y**)). Lipid levels are shown as the amount of picomoles of indicated chain length and lipid species per mg of tissue. For LUAD, unin. n = 22, tumor n = 26; EEC, n = 12; COAD, n = 24; HCC, n = 11; HNSCC, n = 12. Graphs are box plots showing medians (black line) and whiskers of min to max. Uncorrected *p*-values for associations considered significant (α ≤ 0.05) are shown, with values of the predicted mean difference (ANOVA) between unin. and tumor tissue lipid levels shown in parenthesis. The lower limit of *p*- and q-value calculations was set to 1.0 E-15 (Prism), so no values are reported lower than this limit. For *p*-adjusted (q) values corrected for multiple comparisons using the FDR (Q = 0.05) and the two-step controlling procedure of BKY, see Appendix A. For panels (**U**–**Y**), the *y*-axis is in log2 scale. Alterations considered significant (α ≤ 0.05) are in bold text.

**Figure 4 cancers-15-02238-f004:**
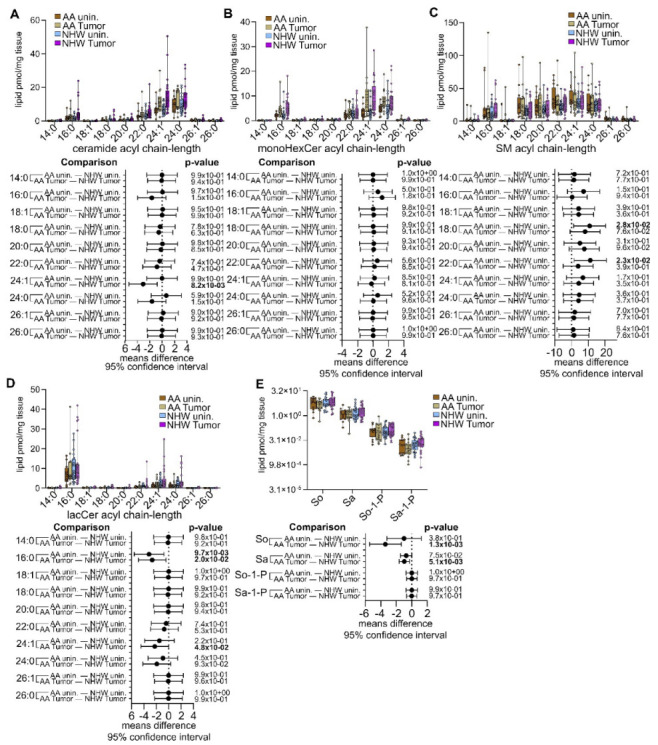
Secondary analysis comparisons of unin. vs. unin. and tumor vs. tumor sphingolipid profiles of self-identified AA and NHW males with LUAD. Data presented in Figure 2A–D,U and Figure 3A–D,U were analyzed to examine differences in the indicated lipids between unin. and tumor tissues of AA and NHW with LUAD. In each panel (**A**–**E**), the upper graphs are box plots showing medians (black line) and whiskers of min to max (top), and the bottom graphs are plots of the 95% family-wise significance and confidence intervals for predicted means difference calculated following ANOVA. Corresponding unadjusted *p*-values (Fisher’s LSD; Prism) for the indicated comparisons are shown to the right of the 95% confidence interval plots. The lower limit of *p*- and q-value calculations was set to 1.0 × 10^−15^ (Prism), so no values are reported lower than this limit. For *p*-adjusted (q) values corrected for multiple comparisons using the FDR (Q = 0.05) and the two-step controlling procedure of BKY, see Appendix A. For panel (**E**), the *y*-axis is in log2 scale. Alterations considered significant (α ≤ 0.05) are in bold text.

**Figure 5 cancers-15-02238-f005:**
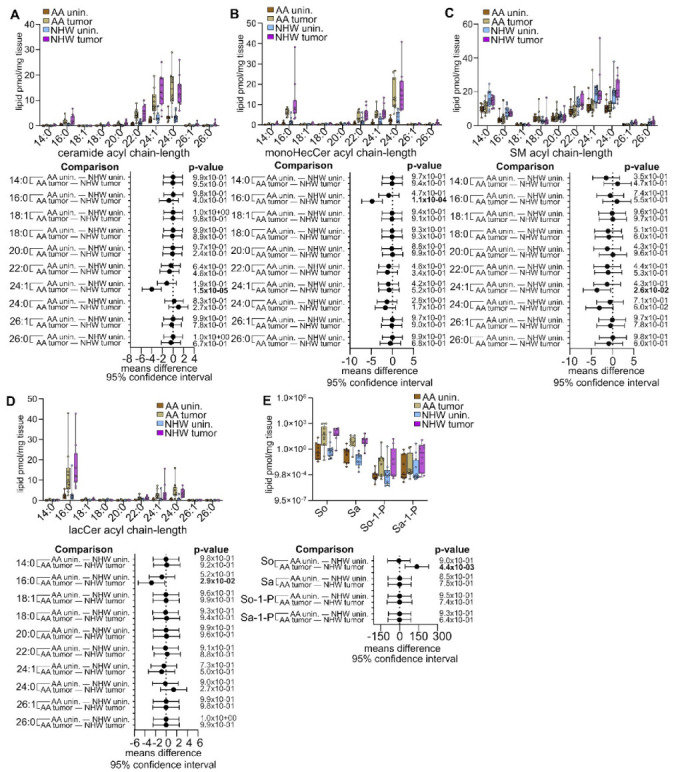
Secondary analysis comparisons of unin. vs. unin. and tumor vs. tumor sphingolipid profiles of self-identified AA and NHW females with EEC. Data presented in Figure 2E–H,V and Figure 3E–H,V were analyzed to examine differences in the indicated lipids between unin. and tumor tissues of self-identified AA and NHW females with EEC. In each panel (**A**–**E**), the upper graphs are box plots showing medians (black line) and whiskers of min to max (top), and the bottom graphs are plots of the 95% family-wise significance and confidence intervals for predicted means difference calculated following ANOVA. Corresponding unadjusted *p*-values (Fisher’s LSD; Prism) for the indicated comparisons are shown to the right of the 95% confidence interval plots. The lower limit of *p*- and q-value calculations was set to 1.0 × 10^−15^ (Prism), so no values are reported lower than this limit. For *p*-adjusted (q) values corrected for multiple comparisons using the FDR (Q = 0.05) and the two-step controlling procedure of BKY, see Appendix A. For panel (**E**), the *y*-axis is in log2 scale. Alterations considered significant (α ≤ 0.05) are in bold text.

**Figure 6 cancers-15-02238-f006:**
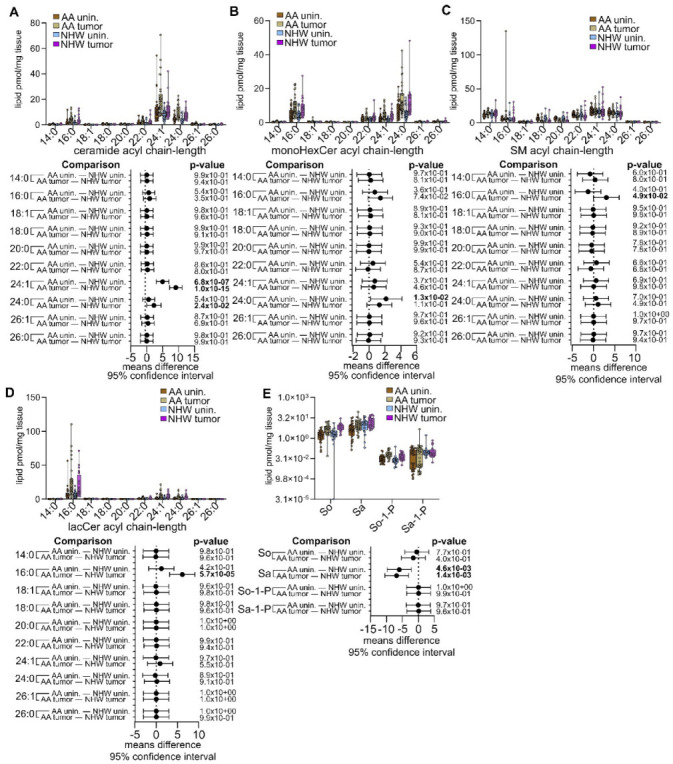
Secondary analysis comparisons of unin. vs. unin. and tumor vs. tumor sphingolipid profiles of self-identified AA and NHW males with COAD. Data presented in Figure 2I–L,W and Figure 3I–L,W were analyzed to examine differences in the indicated lipids between unin. and tumor tissues of self-identified AA and NHW males with COAD. In each panel (**A**–**E**), the upper graphs are box plots showing medians (black line) and whiskers of min to max (top), and the bottom graphs are plots of the 95% family-wise significance and confidence intervals for predicted means difference calculated following ANOVA. Corresponding unadjusted *p*-values (Fisher’s LSD; Prism) for the indicated comparisons are shown to the right of the 95% confidence interval plots. The lower limit of *p*- and q-value calculations was set to 1.0 × 10^−15^ (Prism), so no values are reported lower than this limit. For *p*-adjusted (q) values corrected for multiple comparisons using the FDR (Q = 0.05) and the two-step controlling procedure of BKY, see Appendix A. For panel (**E**), the *y*-axis is in log2 scale. Alterations considered significant (α ≤ 0.05) are in bold text.

**Figure 7 cancers-15-02238-f007:**
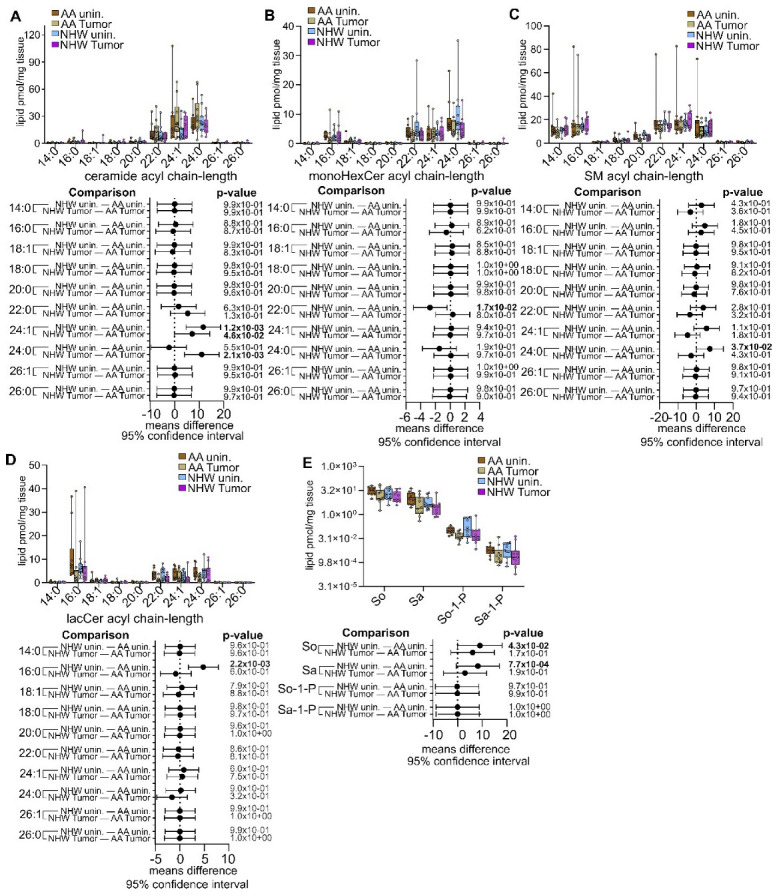
Secondary analysis comparisons of unin. vs. unin. and tumor vs. tumor sphingolipid profiles of self-identified AA and NHW males with HCC. Data presented in Figure 2M–P,X and Figure 3M–P,X were analyzed to examine differences in the indicated lipids between unin. and tumor tissues of self-identified AA and NHW males with HCC. In each panel (**A**–**E**), the upper graphs are box plots showing medians (black line) and whiskers of min to max (top), and the bottom graphs are plots of the 95% family-wise significance and confidence intervals for predicted means difference (means diff.) calculated following ANOVA. Corresponding unadjusted *p*-values (Fisher’s LSD; Prism) for the indicated comparisons are shown to the right of the 95% confidence interval plots. The lower limit of *p*- and q-value calculations was set to 1.0 × 10^−15^ (Prism), so no values are reported lower than this limit. For *p*-adjusted (q) values corrected for multiple comparisons using the FDR (Q = 0.05) and the two-step controlling procedure of BKY, see Appendix A. For panel (**E**), the *y*-axis is in log2 scale. Alterations considered significant (α ≤ 0.05) are in bold text.

**Figure 8 cancers-15-02238-f008:**
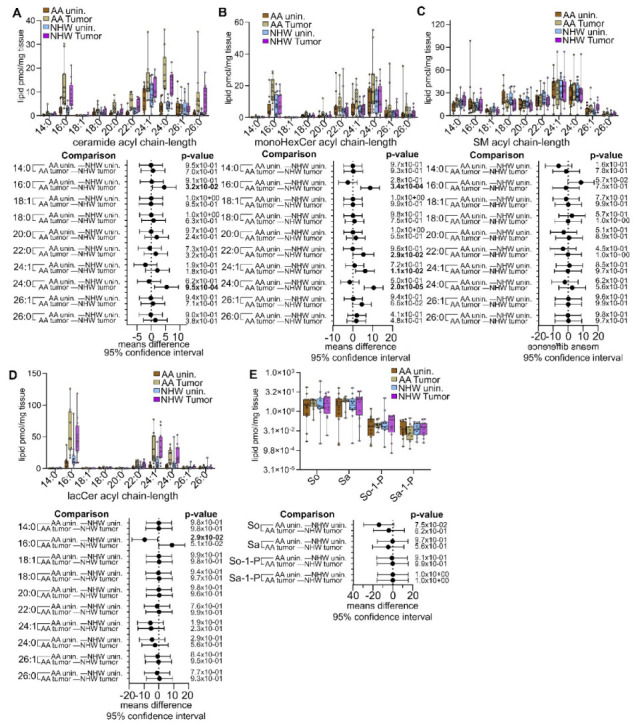
Secondary analysis comparisons of unin. vs. unin. and tumor vs. tumor sphingolipid profiles of self-identified AA and NHW males with HNSCC. Data presented in Figure 2Q–T,Y and Figure 3Q–T,Y were analyzed to examine differences in the indicated lipids between unin. and tumor tissues of self-identified AA and NHW males with HNSCC. In each panel (**A**–**E**), the upper graphs are box plots showing medians (black line) and whiskers of min to max (top), and the bottom graphs are plots of the 95% family-wise significance and confidence intervals for predicted means difference calculated following ANOVA. Corresponding unadjusted *p*-values (Fisher’s LSD; Prism) for the indicated comparisons are shown to the right of the 95% confidence interval plots. The lower limit of *p*- and q-value calculations was set to 1.0 × 10^−15^ (Prism), so no values are reported lower than this limit. For *p*-adjusted (q) values corrected for multiple comparisons using the FDR (Q = 0.05) and the two-step controlling procedure of BKY, see Appendix A. For panel (**E**), the *y*-axis is in log2 scale. Alterations considered significant (α ≤ 0.05) are in bold text.

**Figure 9 cancers-15-02238-f009:**
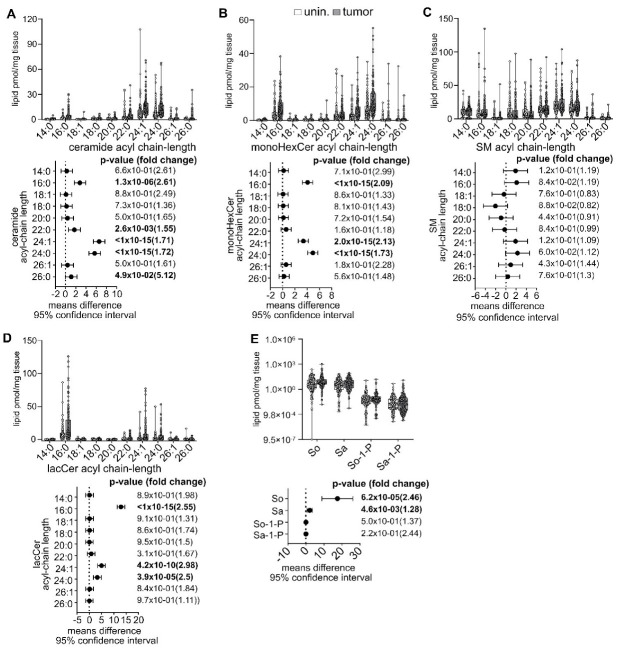
Race-independent pan-cancer analysis of alterations in tumors of subjects with LUAD, COAD, HCC, EEC, and HNSCC. Data from AA and NHW presented in Figure 2 and Figure 3 were pooled by sphingolipid class and acyl chain length and analyzed to examine pan-cancer alterations in human cancers. In each panel (**A**–**E**), the upper graphs are box plots showing medians (black line) and whiskers of min to max (top), and the bottom graphs are plots of the 95% family-wise significance and confidence intervals for predicted means difference calculated following ANOVA. The mean fold-change between unin. tissue and tumor was calculated by dividing the mean lipid level in tumors by the mean lipid level in unin. tissues and shown between parenthesis next to the corresponding unadjusted *p*-values (Fisher’s LSD; Prism). The lower limit of *p*- and q-value calculations was set to 1.0 × 10^−15^ (Prism), so no values are reported lower than this limit. For *p*-adjusted (q) values corrected for multiple comparisons using the FDR (Q = 0.05) and the two-step controlling procedure of BKY, see Appendix A. For panel (**E**), the *y*-axis is in log2 scale. Alterations considered significant (α ≤ 0.05) are in bold text.

**Figure 10 cancers-15-02238-f010:**
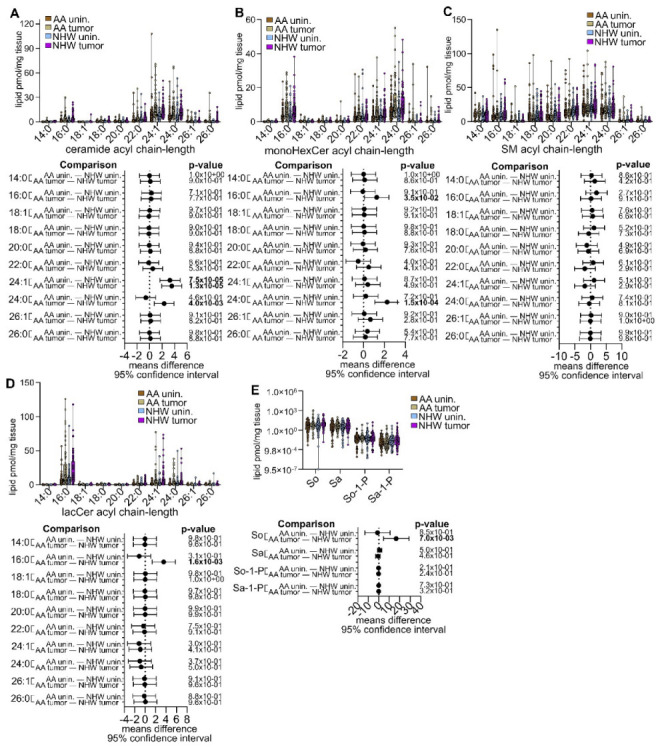
Race-dependent pan-cancer analysis of differences in the unin. and tumor tissues of AA and NHW subjects with LUAD, COAD, HCC, EEC, and HNSCC. Data from AA and NHW presented in Figure 2 and Figure 3 were pooled by race, sphingolipid class, and acyl chain length and analyzed to examine differences between the unin. and tumor tissues of AA and NHW with pan-cancer. In each panel (**A**–**E**), the upper graphs are box plots showing medians (black line) and whiskers of min to max (top), and the bottom graphs are plots of the 95% family-wise significance and confidence intervals for predicted means difference (means diff.) calculated following ANOVA. The mean fold-change between unin. tissue and tumor was calculated by dividing the calculated mean lipid level in tumors by the mean lipid level in unin. tissues and shown between parenthesis next to the corresponding unadjusted *p*-values (Fisher’s LSD; Prism). The lower limit of *p*- and q-value calculations was set to 1.0 × 10^−15^ (Prism), so no values are reported lower than this limit. For *p*-adjusted (q) values corrected for multiple comparisons using the FDR (Q = 0.05) and the two-step controlling procedure of BKY, see Appendix A. For panel (**E**), the *y*-axis is in log2 scale. Alterations considered significant (α ≤ 0.05) are in bold text.

**Table 1 cancers-15-02238-t001:** The characteristics of subjects in lipidomic analysis cohort by disease, race, and sex. *p*-values are for comparisons between the ages of subjects in each group and were analyzed using a two-tailed student’s *t*-test with α = 0.05.

	EEC		LUAD		HCC		HNSCC		COAD	
	AA (n = 10)	NHW (n = 12)	AA (Unin., n = 12; Tumor, n = 14)	NHW (Unin., n = 22; Tumor, n = 26)	AA (n = 10)	NHW (n = 11)	AA (Unin., n = 12; Tumor, n = 11)	NHW (n = 12)	AA (n = 30)	NHW (n = 24)
Sex	Female		Male		Male		Male		Male	
Mean age (SD)	60.7 (9.1)	61.3 (14.4)	60.8 (6.7)	69.6 (9.1)	62.2 (5.5)	64.5 (16.8)	62 (9.5)	58.6 (11.6)	63.5 (13.6)	65.7 (10.5)
Minimum–maximum	49–79	30–82	52–75	47–83	53–73	19–78	39–75	44–78	46–94	41–86
*p*-value	0.9176		0.0018		0.6901		0.4345		0.5214	

## Data Availability

Primary data are not publicly available.

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
