# Peer review of "Lipidomic Profiling Reveals Biological Differences between Tumors of Self-Identified African Americans and Non-Hispanic Whites with Cancer"

_cancers, 2023, doi:10.3390/cancers15082238_

Round 1
Reviewer 1 Report
Authors presented lipidomic profiling between African American and Non-Hispanic whites with cancer and identified a change of lipids between these populations. Authors identified that Sphingolipids may have effect on tumor growth and response to therapy. However, these patients not verified by their ancestry and all are self-reported.
Reviewer 2 Report
Tumor incidence and mortality are disproportionately higher in African Americans (AA). In this study, the authors conducted mass spectrometry analyses of sphingolipids in normal adjacent uninvolved tissues and tumors from self-identified AA and non-Hispanic White (NHW) males with cancers of the lung, colon, liver, and head and neck, and of self-identified AA and NHW females with endometrial cancer. They have identified various sphingolipids are altered in race-specific pat-terns, including differential enrichment of ceramides and glucosylceramides of 24:1 and 24:0 fatty acids in tumors. The results are very interesting and the manuscript was well-prepared. The specific comments are listed below.
1. In the Discussion section, the authors have thoroughly elaborated their findings, such as the higher overall relative levels of the 24:1 and 24:0 fatty acid chain-length ceramides in the tumors of AA and its potential contribution to differences in 24‧16 ceramide ratios. This finding should be properly summarized in the Abstract.
2. In Materials and Methods, the authors stated that tumor sections with greater than 35% tumor enrichment and tissues determined to be disease-free were selected for lipidomic studies. Should the variations in tumor percentage greatly affect the interpretation of lipidomic analyses? The authors should have addressed this issue.
3. No author was identified with affiliation #3. Was that a mistake?
